# The Utility of Intravoxel Incoherent Motion Metrics in Assessing Disability in Relapsing–Remitting Multiple Sclerosis

**DOI:** 10.3390/diagnostics15162113

**Published:** 2025-08-21

**Authors:** Othman I. Alomair, Sami A. Alghamdi, Abdullah H. Abujamea, Salman Aljarallah, Nuha M. Alkhawajah, Mohammed S. Alshuhri, Yazeed I. Alashban, Nyoman D. Kurniawan

**Affiliations:** 1Radiological Sciences Department, College of Applied Medical Sciences, King Saud University, P.O. Box 145111, Riyadh 4545, Saudi Arabia; salghamdi1@ksu.edu.sa (S.A.A.); Yalashban@ksu.edu.sa (Y.I.A.); 2King Salman Centre for Disability Research, Riyadh 11614, Saudi Arabia; 3Department of Radiology and Medical Imaging, King Saud University Medical City & College of Medicine, King Saud University, Riyadh 7805, Saudi Arabia; abujamea@ksu.edu.sa; 4Department of Medicine, College of Medicine, King Saud University, P.O. Box 145111, Riyadh 12372, Saudi Arabia; saljarallah@ksu.edu.sa (S.A.); nalkhawajah@ksu.edu.sa (N.M.A.); 5Radiology and Medical Imaging Department, College of Applied Medical Sciences, Prince Sattam bin Adulaziz University, AlKharj 11942, Saudi Arabia; m.alshuhri@psau.edu.sa; 6Australian Institute for Bioengineering and Nanotechnology, Centre for Advanced Imaging, The University of Queensland, Brisbane, QLD 4072, Australia; nyoman.kurniawan@cai.uq.edu.au

**Keywords:** relapsing–remitting multiple sclerosis (RR-MS), expanded disability status scale (EDSS), intravoxel incoherent motion (IVIM), disease-modifying treatment (DMT), disease duration, number of relapses, mobility assistance

## Abstract

**Background/Objectives**: Quantitative intravoxel incoherent motion (IVIM) imaging, incorporating both diffusion- and perfusion-derived metrics, offers a promising non-invasive approach for assessing tissue microstructure and clinical disability in multiple sclerosis (MS). This study aimed to investigate the correlation and predictive values of the IVIM apparent diffusion coefficient (ADC), true diffusion coefficient (*D*), and perfusion-derived pseudo-diffusion coefficient (*D**) and perfusion fraction (*f*) parameters with disability status, measured using the Expanded Disability Status Scale (EDSS), in relapsing–remitting MS patients. **Methods**: This cross-sectional study retrospectively analyzed MRI data from 197 MS patients. Quantitative IVIM parameters were extracted from scans obtained using a 1.5 T MRI scanner. Clinical data were also obtained, including age, disease duration, number of relapses, disease-modifying therapy (DMT) status, and need for mobility assistance. Bivariate analyses were conducted to compare mean values across subgroups. Pearson correlation was used to examine associations between EDSS score and imaging/clinical variables. Multiple linear regression was applied to identify independent predictors of EDSS score. **Results**: The bivariate analyses revealed that ADC, *D*, *D**, and EDSS values were higher in patients over 50 years old, those with a longer disease duration, and those who required mobility assistance. *f* was higher in females and DMT-treated patients, but it had no effect on EDSS score. Patients with longer disease duration and limited mobility had a higher number of MS lesions and relapses. EDSS score exhibited positive Pearson correlations with ADC, *D*, *D**, the number of MS lesions, and the number of relapses (*p*-value < 0.001). In the multivariate regression analysis, only the number of MS lesions and relapses emerged as independent predictors of EDSS score (*p*-value < 0.001). Other variables, including ADC, *D*, *D**, *f*, age, and disease duration, were not independently associated with EDSS score (*p*-value > 0.05). **Conclusions**: This study demonstrates the utility of IVIM parameters in detecting microstructural alterations associated with MS impairment. Despite relapse frequency and lesion count being the strongest predictors of EDSS score, IVIM metrics showed meaningful clinical correlations. The findings support combining IVIM biomarkers with clinical data for better disability assessment.

## 1. Introduction

Multiple sclerosis (MS) is a chronic autoimmune disease of the central nervous system (CNS) that leads to demyelination and neurodegeneration, resulting in a variety of neurological symptoms such as muscle weakness, visual disturbances, cognitive impairments, and fatigue [1,2,3,4,5,6]. As the disease progresses, MS often leads to increasing disability, affecting patients’ quality of life and their ability to perform daily activities [7,8,9,10,11].

The Expanded Disability Status Scale (EDSS) is the most widely used clinical tool to assess the degree of disability in MS patients, with scores ranging from 0 to 10. A score of 0 indicates no disability, while 10 represents death due to MS. Intermediate scores reflect varying levels of disability, with higher scores indicating more severe impairment in functional systems such as mobility, coordination, vision, and cognitive function [2,9,12]. However, the EDSS score is subjective, and there is ongoing interest in incorporating more objective and sensitive biomarkers, such as MRI-based parameters, to predict disease outcomes more accurately [13,14].

Magnetic resonance imaging (MRI) is the most sensitive non-invasive technique for visualizing and characterizing MS lesions [5,15]. There is growing interest in using MRI-derived measurements as biomarkers in MS clinical trials and treatment monitoring. Several quantitative measures from both conventional and advanced MRI techniques have been examined as potential biomarkers for MS [5,15,16,17,18,19,20]. However, the correlations between EDSS score and a variety of MRI metrics have been inconsistent and modest. Consequently, it is widely recognized that no single parameter derived from MRI can consistently function as a comprehensive imaging biomarker for MS [5,16,21,22,23].

Among advanced MRI techniques, intravoxel incoherent motion (IVIM), derived from diffusion-weighted imaging (DWI) using a bi-exponential model, has emerged as a valuable tool for providing detailed insights into the microstructural changes associated with various stages of MS [20,24,25]. IVIM distinguishes between pure molecular diffusion (*D*), pseudo-diffusion caused by capillary perfusion (*D**), and perfusion fraction (*f*), allowing for a thorough assessment of both diffusion and perfusion within the tissue [20,25,26,27]. This technique allows for the simultaneous measurement of microcirculatory perfusion and tissue diffusion, providing novel insights into the pathophysiological mechanisms underlying neurological diseases, including the development of MS lesions [25,28,29,30].

In contrast, the apparent diffusion coefficient (ADC), derived from traditional DWI, measures the overall diffusivity of water molecules but does not distinguish between molecular diffusion and perfusion influenced by microvascular architecture [5,20,31,32,33,34]. While ADC has been widely used to evaluate demyelination and tissue integrity, particularly in chronic MS lesions, it lacks specificity in characterizing underlying perfusion changes. Therefore, IVIM offers a more detailed assessment of both structural and hemodynamic alterations in brain tissue [20,25].

In our previous work, IVIM parameters demonstrated high sensitivity and specificity in differentiating between various MS lesion types, including non-enhanced, enhanced, and black hole lesions, and normal tissue, highlighting the diagnostic potential of IVIM in MS imaging [20].

The current study aimed to evaluate and correlate IVIM parameters with EDSS scores in patients with MS. By integrating these imaging biomarkers with clinical variables such as relapse history, lesion count, and disease duration, this study sought to elucidate their potential prognostic value in assessing disability progression and enhancing our understanding of MS pathology.

## 2. Materials and Methods

This retrospective cross-sectional study was conducted to evaluate the relationship between IVIM MRI parameters and clinical disability, as measured using the EDSS in patients with MS. The MRI acquisition protocol, image post-processing, and lesion analysis methodology were consistent with our previously published study [20].

### 2.1. Ethical Approval and Participants

This study was approved by King Saud University Medical City’s Institutional Review Board (IRB No. E-23-7517). It began with 224 patients diagnosed with MS [20]. After addressing data completeness and excluding records lacking essential clinical information such as EDSS scores and relapse counts, the final sample used for analysis consisted of 197 patients with complete clinical and imaging datasets. This study’s retrospective nature meant that no informed consent was required. Data were collected from 197 relapsing–remitting MS patients (59 male and 138 female) who had MRI exams as part of their routine clinical evaluation. All patients had a confirmed diagnosis of MS based on the revised McDonald criteria [35,36] and underwent MRI scanning between January 2019 and December 2020. Table 1 presents the demographic details of all patients.

### 2.2. EDSS Evaluation and Relapse History

All participants underwent a comprehensive neurological evaluation that included detailed information about their symptoms, ambulation status, and treatment history. Physical disability was evaluated using the EDSS [2,12,37]. To ensure accuracy and reflect each patient’s clinical condition at the time of imaging, two consultant neurologists obtained the EDSS scores and relapse counts directly from the patients’ medical records. Relapses were defined as new neurological deficits caused by MS that lasted longer than 24 h and were not accompanied by fever or infection. Walking assistance was determined by the type of aid needed for moderate- and long-distance ambulation. McDonald’s 2017 criteria were used to diagnose relapsing–remitting multiple sclerosis [35].

### 2.3. MRI Acquisition Protocol

MRI scans were performed using a 1.5 Tesla GE MRI scanner (GE Healthcare, Waukesha, WI, USA) with an 8-channel phased-array head coil. The imaging protocol included both a standard MS sequence and a diffusion-weighted sequence for IVIM analysis. The IVIM sequence utilized multiple b-values (0, 30, 50, 70, 100, 200, 500, and 1000 s/mm^2^) to generate quantitative maps of diffusion and perfusion parameters, including the ADC, *D*, *D**, and *f*. Detailed imaging parameters were previously reported in Alomair et al. [20].

### 2.4. Image Post-Processing

IVIM parameters were extracted using the IB Diffusion™ plug-in (version 21.12, Imaging Biometrics, Elm Grove, WI, USA) integrated into the OsiriX MD platform [20,38]. The segmented IVIM model was employed, with a b inflection point set at 200 s/mm^2^. *D** was calculated using b-values below this threshold, while *D* was calculated using b-values above it. Perfusion fraction (*f*) was estimated as the ratio of the perfusion-related signal to the total signal decay [20].

All MS lesions were identified on conventional weighted images and confirmed by two neuroradiologists with over 10 years of experience. The number of lesions was recorded for each patient. Regions of interest (ROIs) were manually drawn on post-contrast 2D T_1_-weighted images (WI) using ITK-SNAP [28], guided by registered 2D T_2_WI images to ensure accurate localization of MS lesions [39,40,41]. ROIs were placed in representative lesions, avoiding areas of partial volume effects or artifacts. The same neuroradiologist performed the ROI analysis for all patients to maintain consistency. Further details are described in our previous study [20].

### 2.5. Data Analysis

All statistical analyses were performed using IBM SPSS Statistics version 26.0 (IBM Corp., Armonk, NY, USA). Descriptive statistics (mean values, standard deviations, frequencies, and percentages) were computed for demographic and clinical variables. Student’s *t*-test was used for comparisons between two groups, and one-way ANOVA followed by Tukey’s post hoc test was applied for multiple comparisons. Corresponding effect sizes were calculated to quantify the magnitude of group differences: Cohen’s d was used for two-group comparisons, while partial eta squared (η^2^) was used for comparisons involving more than two groups for both MRI and clinical outcome variables.

Pearson correlation coefficients were calculated to assess the strength and direction of linear relationships between quantitative variables. Multiple linear regression analysis was used to identify significant independent predictors of EDSS scores, with imaging and clinical variables input as the independent predictors. A *p*-value ≤ 0.05 was considered statistically significant in all analyses.

## 3. Results

### 3.1. Descriptive Statistics

Among the 197 MS patients included in this study, 57.4% were between 31 and 50 years old, and 70.1% were female. A total of 44.2% had a disease duration of more than five years, 73.1% were receiving DMT, and 10.2% required mobility assistance. The mean quantitative MRI metrics are ADC = 1.10 (±0.14) × 10^−3^ mm^2^/s, *D* = 1.04 (±0.14) × 10^−3^ mm^2^/s, *D** = 1.27 (±0.16) × 10^−3^ mm^2^/s, and *f* = 0.06 (±0.02) (%). Clinically, the mean EDSS score was 2.25 (±1.91), with an average of 11.73 (±8.28) lesions and 2.65 (±2.03) relapses (Table 1).

Representative conventional MRI sequences, together with the corresponding IVIM parametric maps from a patient with RR-MS in our study, are shown in Figure 1.

### 3.2. Bivariate Analysis

This section presents the results of the bivariate analysis, as shown in Table 2 and Table 3, comparing the mean values of the clinical outcome variables—ADC, *D*, *D**, *f*, EDSS score, number of MS lesions, and number of relapses—across key patient subgroups defined by age group, sex, disease duration, DMT status, and mobility support.

#### 3.2.1. Association Between IVIM Parameters and Clinical Variables

The bivariate analysis showed significant associations between the quantitative diffusion- and perfusion-derived IVIM metrics (ADC, *D*, *D**, *f*) and several clinical characteristics (Table 2). Participants aged > 50 years exhibited higher ADC (*p* < 0.001), *D* (*p* < 0.001), and *D** (*p* = 0.002) values than the other two groups. Similarly, participants with disease duration > 5 years had significantly elevated ADC, *D*, and *D** values (all *p* < 0.001). Patients requiring mobility assistance displayed higher ADC (*p* = 0.001), *D* (*p* = 0.002), and *D** (*p* = 0.004) than those who were independently ambulatory.

The *f* value did not differ by age, disease duration, or mobility status but was greater in females than males (*p* = 0.008) and in individuals receiving DMT (*p* = 0.015). In contrast, *D* was the only diffusion coefficient significantly affected by DMT use (*p* = 0.036), while ADC and *D** showed no treatment-related differences. These findings indicate that advanced age, longer disease duration, and impaired mobility are consistently associated with increased diffusion and pseudo-diffusion metrics, whereas *f* varies mainly with sex and treatment status.

#### 3.2.2. Disease Burden and Disability Measures

Table 3 summarizes the relationships between disease-burden indicators (lesion count and relapse frequency) and EDSS score across clinical subgroups. Patients with a disease duration > 5 years had markedly more MS lesions (*p* < 0.001), more relapses (*p* < 0.001), and higher EDSS scores (*p* < 0.001) than those with a shorter disease course.

Likewise, individuals requiring mobility assistance demonstrated both a greater lesion burden (*p* < 0.001) and higher EDSS scores (*p* < 0.001), although their relapse frequency did not differ significantly from independently mobile patients (*p* = 0.075).

Participants aged >50 years showed a higher mean EDSS score (*p* = 0.007) but did not differ in lesion or relapse counts. DMT-treated patients experienced more relapses than untreated patients (*p* < 0.001); however, their lesion counts and EDSS scores were comparable. No significant differences in lesion burden, relapse rate, or EDSS score were observed between sexes. Collectively, these data confirm that prolonged disease, impaired mobility, and older age contribute to greater neurological disability, while treatment status primarily influences relapse frequency.

### 3.3. Correlation Analysis

Table 4 presents the correlations between EDSS scores and various imaging and clinical variables. Statistically positive correlations (*p*-value < 0.001) were observed between EDSS score and the following parameters: ADC, *D*, *D**, number of MS lesions, and number of relapses. These findings indicate that as EDSS score increases, there is a corresponding increase in these imaging-derived diffusion and perfusion measures, lesion burden, and relapse frequency. No significant correlation was found between EDSS score and *f*.

### 3.4. Group Comparison According to EDSS Level

Table 5 shows the comparison of outcome variables across EDSS scores <3.0 and ≥3.0. Significantly higher mean values of ADC, *D*, *f*, and number of lesions were found in the higher EDSS group (≥3.0). *D** did not differ significantly between the two EDSS levels.

### 3.5. Multivariate Regression Analysis

Multiple linear regression analyses were conducted to evaluate the independent predictors of EDSS score using various combinations of imaging-derived parameters (ADC, *D*, *D**, *f*), along with age, disease duration, number of MS lesions, and number of relapses (Table 6, Table 7, Table 8 and Table 9). Across all models, the number of MS lesions and the number of relapses were consistently identified as significant independent predictors of EDSS score (*p* < 0.05), with regression coefficients ranging from 0.072 to 0.146.

These results suggest that higher lesion burden and relapse frequency are associated with greater disability levels. In contrast, the imaging parameters (ADC, *D*, *D**, *f*), age, and disease duration did not exhibit significant independent associations with EDSS score in any model. Each model demonstrated statistical significance (F-values ranging from 6.36 to 10.27, all *p* < 0.001), with R^2^ values between 0.232 and 0.238, indicating that approximately 23–24% of the variance in EDSS scores is explained by these predictors.

Table 10 shows the multivariable model examining predictors of EDSS score after simultaneously adjusting for age, IVIM-derived diffusion and perfusion metrics (ADC, *D*, *D**, *f*), number of lesions, relapse counts, and disease duration. The overall model fit was significant (F = 6.36, *p* < 0.0001) and accounted for 23.8% of the variance in EDSS score; however, only lesion count (β = 0.072 ± 0.016, *p* < 0.001) and number of relapses (β = 0.138 ± 0.063, *p* = 0.031) emerged as independent determinants of disability: each additional lesion raised the EDSS score by ≈0.07 points and each relapse by ≈0.14 points. In contrast, age, disease duration, and all IVIM parameters did not demonstrate significant associations with EDSS score after accounting for lesion and relapse counts.

## 4. Discussion

This study investigated the relationships between quantitative MRI parameters derived from diffusion-weighted imaging (ADC, *D*, *D**, and *f*) and clinical variables, including age, disease duration, mobility status, number of MS lesions, and relapse frequency, in patients with MS. Bivariate analysis revealed a number of clinically meaningful associations, highlighting the potential role of both MRI biomarkers and clinical characteristics in understanding MS-related disability. The main finding of this study is that IVIM parameters—specifically, ADC, *D*, and *D**—are significantly associated with EDSS scores in MS patients. These parameters remained relevant in multivariable models alongside lesion count and relapse frequency, highlighting their potential as imaging biomarkers for assessing disability in MS.

### 4.1. MRI Biomarkers and Clinical Associations

Diffusion-based parameters, including ADC, *D*, and *D**, showed significantly higher mean values in patients older than 50 years, those with disease duration greater than five years, and those requiring support for mobility.

These findings suggest that water diffusivity and perfusion-related diffusion characteristics may reflect cumulative neurological damage due to aging and disease progression. Conversely, the *f* parameter, which estimates perfusion fraction, was significantly elevated in female patients and those receiving DMT, indicating possible sex-related perfusion differences and a treatment-related influence on vascular features in MS lesions.

Furthermore, a published study by Alomair et al. demonstrated that ADC, *D*, and *D** were highly sensitive in distinguishing MS lesions, particularly black holes, from normal white matter, with high diagnostic accuracy, indicating the robustness of these IVIM parameters as non-invasive imaging biomarkers for MS lesion characterization [20].

### 4.2. Disability Scores and Clinical/Demographic Correlates

The findings from our bivariate analyses reveal a statistically significant increase in EDSS scores among patients over 50 years of age [42], those with disease duration exceeding five years, and individuals requiring support for mobility. These results reinforce the association between clinical characteristics and functional disability in MS. Such observations are consistent with prior studies; for example, Rzepiński et al. [43] reported that longer disease duration correlates with higher EDSS scores and earlier attainment of disability milestones such as EDSS 3 and 6 [44].

Moreover, the need for mobility support was found to be strongly associated with increased disability, consistent with established clinical understanding that reduced ambulation reflects more severe neurological impairment [45]. Interestingly, our study found no significant differences in EDSS scores between male and female patients or in relation to DMT treatment. This observation suggests that while MS incidence is higher in females [46,47], the progression of disability, as measured using the EDSS score, does not significantly differ between sexes. Furthermore, the lack of association between DMT treatment and EDSS scores may suggest variability in treatment response or disease stage among patients, highlighting the need for personalized therapeutic approaches.

### 4.3. EDSS Stratification Threshold

The use of an EDSS threshold of <3 versus ≥3 was based on a previously established approach [48,49], identifying EDSS ≥3 as a clinically meaningful marker of transition from minimal to moderate disability. This threshold reflects a pivotal stage in MS progression where functional limitations become more apparent, particularly in ambulation and motor performance. The study by Coll et al. [49], which used deep learning models to decode clinical disability from MRI features, supports this cut-off by demonstrating that patients with EDSS ≥ 3 exhibit more distinct and widespread structural brain changes. These findings validate the clinical utility of this stratification in capturing disease impact and justify its use in assessing associations with imaging and clinical parameters in MS research.

### 4.4. Correlation and Predictive Analyses

Pearson correlation analysis revealed positive associations between EDSS score and the imaging metrics ADC, *D*, and *D**, indicating that higher diffusion values correlate with increased disability. However, *f* showed no significant correlation with EDSS score, suggesting that *f* alone may not serve as a strong indicator of disability severity in MS. Additionally, significant correlations were found between EDSS score and the number of MS lesions and the number of relapses. These associations further support the role of relapse activity and lesion burden in contributing to disease progression.

Multiple regression models confirmed these trends. In the fully adjusted model that included all imaging and clinical variables (Table 10), relapse frequency and number of lesions remained the only independent predictors of disability (β = 0.072 and β = 0.138, respectively; both *p* < 0.05), together accounting for roughly one quarter of the variance in EDSS score (R^2^ = 0.238, F = 6.36, *p* < 0.0001). Age, disease duration, and each IVIM parameter lost statistical significance once relapse frequency and number of lesions were entered. These results mirror earlier reports by Dworkin et al. [50], Tomassini et al. [51], and Popescu et al. [52], which identified lesion counts and relapse history as robust determinants of disability progression in MS.

Although our regression models identified statistically significant associations, the R^2^ values (approximately 24%) indicate that a substantial portion of the variability in EDSS scores remains unexplained. This reflects the complex and multifactorial nature of disability in multiple sclerosis and highlights the limited predictive power of our model, which included only IVIM-derived metrics and basic clinical variables. To address this limitation, it is essential to consider the integration of additional clinical, neuropsychological, and advanced neuroimaging predictors in future investigations. For instance, a recent longitudinal study by Lopez-Soley et al. [53] demonstrated that diffusion tensor imaging (DTI) metrics, particularly fractional anisotropy in white matter tracts, were predictive of future disability in patients with relapsing–remitting multiple sclerosis. Despite the modest explanatory power of their models, the study supports the value of incorporating multiple structural and functional imaging biomarkers alongside clinical assessments to enhance prediction accuracy and capture the heterogeneous nature of disability progression in multiple sclerosis.

The observed associations between IVIM-derived parameters and EDSS scores underscore the potential utility of these imaging metrics in clinical management of MS. Specifically, ADC, *D*, and *D** may serve as supportive markers for monitoring microstructural brain alterations that are not evident on conventional MRI. These metrics could aid in identifying patients at risk of early functional decline or in evaluating therapeutic response, particularly in cases where clinical symptoms and conventional imaging findings are incongruent. Incorporating IVIM into longitudinal imaging protocols may enhance sensitivity to subtle disease progression, offering clinicians additional quantitative tools to inform prognosis and guide individualized treatment strategies.

### 4.5. Limitations

Despite the promising results, this study has several limitations. First, it was conducted at a single hospital, which limits the generalizability of the findings to other settings, especially those with different patient demographics or clinical practices. Additionally, the cross-sectional design and the inclusion of only a single time point restrict the ability to assess long-term changes in IVIM MRI parameters and EDSS scores over time. Longitudinal studies with multiple time points are needed to track disease progression and the relationship between MRI biomarkers and clinical outcomes more comprehensively [54]. Longitudinal studies are particularly valuable for capturing dynamic changes in both imaging biomarkers and clinical disability over time, allowing for the identification of predictive trajectories and causal inferences. Such designs would enhance the temporal validity of IVIM metrics in relation to MS progression and treatment response.

Additionally, this study did not stratify patients according to MS subtype (e.g., enhancing, non-enhancing, black holes), which may have influenced the observed variability in clinical and imaging correlations [20]. Future research should incorporate MS subtypes to provide a more nuanced understanding of disease progression and treatment response across the MS spectrum.

## 5. Conclusions

This study provides preliminary evidence that IVIM MRI parameters, particularly ADC, *D*, and *D**, are associated with clinical disability in patients with multiple sclerosis. While lesion burden and relapse frequency were the strongest independent predictors of EDSS scores in our multivariate model, IVIM-derived diffusion metrics showed statistically significant, though modest, associations with variables such as age, disease duration, and mobility status. These findings suggest that IVIM metrics may have supportive value in assessing MS-related microstructural changes when used alongside established clinical measures. However, given the limited explanatory power of our models, further validation is necessary. Future studies should adopt longitudinal designs, stratify patients by lesion subtype and MS phenotype, and incorporate complementary neuroimaging and clinical predictors to enhance the prognostic utility of IVIM MRI in personalized disease monitoring and management.

## Figures and Tables

**Figure 1 diagnostics-15-02113-f001:**
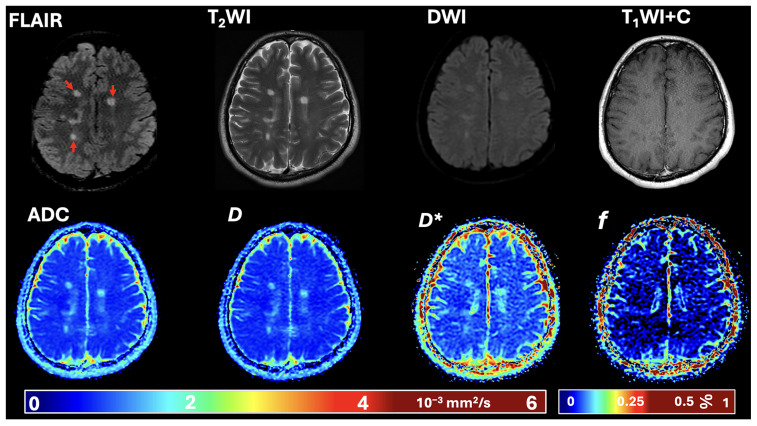
Representative MRI examination in an RR-MS patient. The axial images depict, from top left to right, FLAIR (red arrows indicate MS lesions), T_2_-weighted image, diffusion-weighted image, and post-contrast T_1_-weighted sequences; from bottom left to right are the parametric maps of the apparent diffusion coefficient (ADC), diffusion coefficient (*D*), pseudo-diffusion coefficient (*D**), and perfusion fraction (*f*).

**Table 1 diagnostics-15-02113-t001:** Descriptive statistics of demographics and outcome variables.

Demographics and Outcome Variables	*N* (%)	Mean ± SD
Age groups (in years)		
≤30	66 (33.5%)	
31 to 50	113 (57.4%)	
>50	18 (9.1%)	
Gender		
Male	59 (29.9%)	
Female	138 (70.1%)	
Disease duration (in years)		
<2	38 (19.3%)	
2 to 5	72 (36.5%)	
>5	87 (44.2%)	
DMT		
Treated	144 (73.1%)	
Not treated	53 (26.9%)	
Mobility		
Supported	20 (10.2%)	
Not supported	177 (89.8%)	
ADC		1.10 ± 0.14
*D*		1.04 ± 0.14
*D**		1.30 ± 0.16
*f*		0.06 ± 0.02
EDSS		2.25 ± 1.91
No. of MS lesions	2310 for all RR-MS patients	11.7 ± 8.3
Number of relapses	463 for all RR-MS patients	2.65 ± 2.03

Note: *N* = number, SD = standard deviation, DMT = disease-modifying treatment, EDSS = Expanded Disability Status Scale, ADC = apparent diffusion coefficient, *D* = diffusion coefficient, *D** = pseudo-diffusion coefficient, *f* = perfusion fraction, RR-MS = relapsing–remitting multiple sclerosis, and the units for ADC, *D*, and *D** are 10^−3^ mm^2^/s.

**Table 2 diagnostics-15-02113-t002:** Comparison of mean values of MRI outcome variables in relation to demographic and other study variables.

Study Variables	ADC		*D*		*D**		*f*	
	Mean ± SD	*p*-Value and Effect Size	Mean ± SD	*p*-Value and Effect Size	Mean ± SD	*p*-Value and Effect Size	Mean ± SD	*p*-Value and Effect Size
Age groups (in years)								NS
≤30	1.06 ± 0.15	<0.001 and 0.080	1.01 ± 0.15	<0.001 and 0.080	1.23 ± 0.17	0.002 and 0.060	0.06 ± 0.01	
31–50	1.10 ± 0.13		1.04 ± 0.13		1.27 ± 0.15		0.06 ± 0.02	
>50	1.22 ± 0.11		1.16 ± 0.10		1.37 ± 0.13		0.06 ± 0.01	
Gender		NS		NS		NS		
Male	1.09 ± 0.14		1.04 ± 0.13		1.24 ± 0.15		0.054 ± 0.01	0.008 and −0.417
Female	1.10 ± 0.15		1.04 ± 0.14		1.28 ± 0.16		0.061 ± 0.02	
Disease duration (in years)								NS
<2	0.99 ± 0.11	<0.001 and 0.185	0.94 ± 0.11	<0.001 and 0.182	1.18 ± 0.15	<0.001 and 0.117	0.06 ± 0.02	
2 to 5	1.08 ± 0.12		1.03 ± 0.12		1.25 ± 0.15		0.05 ± 0.02	
>5	1.16 ± 0.14		1.10 ± 0.14		1.32 ± 0.15		0.06 ± 0.01	
DMT		NS				NS		
Treated	1.11 ± 0.15		1.05 ± 0.14	0.036 and 0.339	1.27 ± 0.16		0.057 ± 0.01	0.015 and −0.394
Not treated	1.06 ± 0.13		1.01 ± 0.12		1.25 ± 0.16		0.063 ± 0.02	
Mobility								NS
Supported	1.20 ± 0.17	0.001 and 0.781	1.14 ± 0.16	0.002 and 0.784	1.37 ± 0.19	0.004 and 0.676	0.057 ± 0.01	
Not supported	1.08 ± 0.14		1.03 ± 0.13		1.25 ± 0.15		0.059 ± 0.02	

Note: NS = not significant; SD = standard deviation; DMT = disease-modifying treatment; ADC = apparent diffusion coefficient; *D* = diffusion coefficient; *D** = pseudo-diffusion coefficient; *f* = perfusion fraction; units for ADC, *D*, and *D** are 10^−3^ mm^2^/s.

**Table 3 diagnostics-15-02113-t003:** Comparison of mean values of clinical outcome variables in relation to demographic and other study variables.

Study Variables	Outcome Variables					
	No. of MS Lesions		No. of Relapses		EDSS	
	Mean ± SD	*p*-Value and Effect Size	Mean ± SD	*p*-Value and Effect Size	Mean ± SD	*p*-Value and Effect Size
Age groups (in years)		NS		NS		
≤30	10.68 ± 8.05		2.65 ± 2.45		1.97 ± 1.70	
31–50	12.31 ± 8.57		2.60 ± 1.52		2.20 ± 1.87	0.007 and 0.050
>50	11.89 ± 7.30		3.0 ± 3.36		3.59 ± 2.48	
Gender		NS		NS		NS
Male	12.07 ± 8.26		2.59 ± 2.33		2.27 ± 2.0	
Female	11.58 ± 8.32		2.67 ± 1.88		2.24 ± 1.8	
Disease duration (in years)						
<2	8.53 ± 7.95	<0.001 and 0.078	1.50 ± 0.80	<0.001 and 0.150	1.55 ± 1.4	<0.001 and 0.126
2 to 5	10.46 ± 8.28		2.41 ± 2.10		1.70 ± 1.4	
>5	14.17 ± 7.80		3.54 ± 2.05		3.01 ± 2.2	
DMT		NS		<0.001 and 0.651		NS
Treated	11.44 ± 7.46		3.01 ± 2.23		2.35 ± 2.0	
Not treated	12.49 ± 10.23		1.74 ± 0.85		1.95 ± 1.7	
Mobility				NS		
Supported	18.45 ± 10.15	<0.001 and 0.918	3.57 ± 2.95		6.58 ± 0.70	<0.001 and 4.158
Not supported	10.97 ± 7.72		2.57 ± 1.92		1.80 ± 1.40	

Note: NS = not significant; SD = standard deviation; DMT = disease-modifying treatment; EDSS = Expanded Disability Status Scale.

**Table 4 diagnostics-15-02113-t004:** Correlation between EDSS score and other clinical outcome variables.

IVIM and Clinical Variables	EDSS	
	Pearson’s Correlation	*p*-Value
ADC	0.360	<0.001
*D*	0.368	<0.001
*D**	0.283	<0.001
*f*	−0.106	NS
No. of MS lesions	0.372	<0.001
Number of relapses	0.259	0.001

Note: NS = not significant; EDSS = Expanded Disability Status Scale; ADC = apparent diffusion coefficient; *D* = diffusion coefficient; *D** = pseudo-diffusion coefficient; *f* = perfusion fraction; the units for ADC, *D*, and *D** are 10^−3^ mm^2^/s.

**Table 5 diagnostics-15-02113-t005:** Comparison of mean values of ADD, IVIM, and clinical outcome variables across categories of EDSS score.

EDSS	MRI and Clinical Outcome Variables									
	ADC		*D*		*D**		*f*		No. of Lesions	
	Mean ± SD	*p*-Value and Effect Size	Mean ± SD	*p*-Value and Effect Size	Mean ± SD	*p*-Value and Effect Size	Mean ± SD	*p*-Value and Effect Size	Mean ± SD	*p*-Value and Effect Size
<3.0 ≥3.0	1.08 ± 0.13 1.15 ± 0.17	0.001 and −0.504	1.02 ± 0.13 1.10 ± 0.16	0.001 and −0.539	1.25 ± 0.15 1.30 ± 0.18	0.067 and −0.288	0.06 ± 0.02 0.05 ± 0.01	0.046 and 0.314	9.9 ± 6.80 16.1 ± 9.80	<0.001 and −0.785

Note: EDSS = Expanded Disability Status Scale; ADC = apparent diffusion coefficient; *D* = diffusion coefficient; *D** = pseudo-diffusion coefficient; *f* = perfusion fraction; units for ADC, *D*, and *D** are 10^−3^ mm^2^/s.

**Table 6 diagnostics-15-02113-t006:** Relationship between EDSS score and ADC with other clinical variables (using multiple regression analysis).

Variables	β Coefficients	Standard Error	t-Value	*p*-Value	95% CI for β Coefficient
Constant	−0.488	1.079	−0.452	NS	−2.62, 1.64
Age	0.025	0.015	1.651	NS	−0.005, 0.054
ADC	0.334	1.052	0.317	NS	−1.74, 2.41
Number of MS lesions	0.074	0.016	4.769	<0.001	0.04, 0.10
Number of relapses	0.146	0.63	2.326	0.021	0.02, 0.27
Disease duration	0.010	0.028	0.374	NS	−0.4, 0.06

R-square = 0.232, F = 10.0; *p* < 0.001; Note: NS = not significant; EDSS = Expanded Disability Status Scale; ADC = apparent diffusion coefficient; units for ADC are 10^−3^ mm^2^/s.

**Table 7 diagnostics-15-02113-t007:** Relationship between EDSS score and *D* with other clinical variables (using multiple regression analysis).

Variables	β Coefficients	Standard Error	t-Value	*p*-Value	95% CI for β Coefficient
Constant	−0.614	1.058	−580	NS	−2.70, 1.47
Age	0.024	0.015	1.633	NS	−005, 0.05
*D*	0.499	1.078	0.463	NS	−1.63, 2.63
Number of MS lesions	0.073	0.015	4.755	<0.001	0.04, 0.10
Number of relapses	0.145	0.063	2.319	0.022	0.02, 0.27
Disease duration	0.010	0.028	0.346	NS	−0.045, 0.06

R-square = 0.232; F = 10.04; *p* < 0.001; Note: NS = not significant; EDSS = Expanded Disability Status Scale; *D* = diffusion coefficient; units for *D* are 10^−3^ mm^2^/s.

**Table 8 diagnostics-15-02113-t008:** Relationship between EDSS score and *D** with other clinical variables (using multiple regression analysis).

Variables	β Coefficients	Standard Error	t-Value	*p*-Value	95% CI for β Coefficient
Constant	0.099	1.059	0.093	NS	−1.99, 2.19
Age	0.027	0.015	1.772	NS	−0.003, 0.06
*D**	−0.279	0.888	−0.314	NS	−2.03, 1.47
Number of MS lesions	0.077	0.015	5.063	<0.001	0.05, 0.11
Number of relapses	0.144	0.063	2.287	0.023	0.02, 0.27
Disease duration	0.013	0.027	0.483	NS	−0.04, 0.07

R-square = 0.232; F = 10.01; *p* < 0.001; Note: NS = not significant; EDSS = Expanded Disability Status Scale; *D** = pseudo-diffusion coefficient; units for and *D** are 10^−3^ mm^2^/s.

**Table 9 diagnostics-15-02113-t009:** Relationship between EDSS score and *f* with other clinical variables (using multiple regression analysis).

Variables	β Coefficients	Standard Error	t-Value	*p*-Value	95% CI for β Coefficient
Constant	0.219	0.655	0.335	NS	−1.07, 1.51
Age	0.028	0.015	1.885	NS	−0.001, 0.06
*f*	−7.546	7.205	−1.047	NS	−21.77, 6.68
Number of MS lesions	0.074	0.015	5.079	<0.001	0.04, 0.10
Number of relapses	0.138	0.063	2.198	0.029	0.014, 0.26
Disease duration	0.011	0.027	0.407	NS	−0.04, 0.06

R-square = 0.236; F = 10.27; *p* < 0.0001; Note: NS = not significant; EDSS = Expanded Disability Status Scale; *f* = perfusion fraction.

**Table 10 diagnostics-15-02113-t010:** Relationship between EDSS score and other clinical variables (using multiple regression analysis).

Variables	β Coefficients	Standard Error	t-Value	*p*-Value	95% CI for β Coefficient
Constant	−0.184	1.136	−0.162	NS	−2.43, 2.06
Age	0.027	0.015	1.780	NS	−0.003, 0.06
ADC	1.262	5.674	0.222	NS	−9.94, 12.46
*D*	0.007	5.472	0.001	NS	−10.80, 10.81
*D**	−0.831	2.502	−0.332	NS	−5.77, 4.11
*f*	−5.211	12.851	−0.406	NS	−30.59, 20.16
Number of MS lesions	0.072	0.016	4.506	<0.001	0.041, 0.10
Number of relapses	0.138	0.063	2.179	0.031	0.013, 0.26
Disease duration	0.008	0.028	0.284	NS	−0.05, 0.06

R-square = 0.238; F = 6.36; *p* < 0.0001; NS = not significant; EDSS = Expanded Disability Status Scale; ADC = apparent diffusion coefficient; *D* = diffusion coefficient; *D** = pseudo-diffusion coefficient; *f* = perfusion fraction; units for ADC, *D*, and *D** are 10^−3^ mm^2^/s.

## Data Availability

The data presented in this study are available on request from the corresponding author, as they are currently being used in another clinical experiment.

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
