# Peer review of "The Utility of Intravoxel Incoherent Motion Metrics in Assessing Disability in Relapsing–Remitting Multiple Sclerosis"

_diagnostics, 2025, doi:10.3390/diagnostics15162113_

Round 1

Reviewer 1 Report

Comments and Suggestions for Authors

This manuscript explores the utility of intravoxel incoherent motion (IVIM) imaging metrics in assessing disability in patients with relapsing-remitting multiple sclerosis (RR-MS). Researchers investigated correlations between IVIM parameters (apparent diffusion coefficient (ADC), true diffusion coefficient (D), pseudo-diffusion coefficient (D*), and perfusion fraction (f)) and the Expanded Disability Status Scale (EDSS) scores, a common measure of MS-related disability. The study analyzed MRI data and clinical information from 197 RR-MS patients, finding that while IVIM parameters showed associations with age, disease duration, and mobility, lesion count and relapse frequency were the strongest independent predictors of EDSS scores. The authors conclude that combining IVIM biomarkers with clinical data holds promise for a more comprehensive assessment of MS impairment.

This is a well-written retrospective study. IVIM is a valuable imaging technique. But I have some concerns that do not allow me to recommend this paper for publication:

1. The reported correlations and regression coefficients are weak in magnitude. The bivariate analysis, which examines the direct relationship between two variables, shows statistically significant but weak-to-moderate correlations between the EDSS score and other variables. The strongest correlation reported is between the EDSS score and the number of MS lesions (Pearson's r = 0.372). In statistical terms, correlation coefficients in the 0.2 to 0.4 range are typically considered to indicate a weak to moderate relationship. So, your observation should be interpreted with great caution. Further, the models explained approximately 23-24% of the variance in EDSS scores, with R-squared values ranging from 0.232 to 0.238. This means that over 75% of the variability in patient disability is not explained by the combination of variables used in this study (age, disease duration, IVIM metrics, lesion count, and relapses)! Are you sure this is meaningful for report? I am not sure. I recommend the authors add some imaging, clinical, and neuropsychological predictors to improve their models.

1. Please avoiding to replcate your findings such as the first paragraph of results. These results replicated in the body text and results (section 3.1.).

2. P values alone not reflective of how measure effect. Kindly report appropriate effect size along with the p value reports.

Author Response

Comments 1: The reported correlations and regression coefficients are weak in magnitude. The bivariate analysis, which examines the direct relationship between two variables, shows statistically significant but weak-to-moderate correlations between the EDSS score and other variables. The strongest correlation reported is between the EDSS score and the number of MS lesions (Pearson's r = 0.372). In statistical terms, correlation coefficients in the 0.2 to 0.4 range are typically considered to indicate a weak to moderate relationship. So, your observation should be interpreted with great caution.

Further, the models explained approximately 23-24% of the variance in EDSS scores, with R-squared values ranging from 0.232 to 0.238. This means that over 75% of the variability in patient disability is not explained by the combination of variables used in this study (age, disease duration, IVIM metrics, lesion count, and relapses)! Are you sure this is meaningful for report? I am not sure. I recommend the authors add some imaging, clinical, and neuropsychological predictors to improve their models.

Comments 2: I recommend the authors add some imaging, clinical, and neuropsychological predictors to improve their models.

Response to Comments 1 and 2:

Thank you for your thoughtful and constructive comments. We acknowledge that the reported correlations and regression coefficients indicate weak to moderate associations, with R² values around 24%, suggesting that a substantial proportion of variability in EDSS scores remains unexplained. This limitation highlights the complex and multifactorial nature of disability in multiple sclerosis and the challenges of modeling its clinical progression using a limited set of predictors.

In response, we have expanded the end of Section 4.4 (Correlation and Predictive Analyses) in the Discussion to address the restricted explanatory power of our regression models and to emphasize the importance of including additional clinical, neuropsychological, and advanced imaging variables in future studies. Specifically, we included reference to recent studies supporting the use of complementary neuroimaging modalities such as diffusion tensor imaging (DTI). For instance, a longitudinal study by Lopez-Soley et al. (2023) demonstrated that fractional anisotropy in brain white matter tracts was predictive of EDSS progression in patients with relapsing–remitting MS. These findings support the need to integrate multimodal imaging markers into more comprehensive predictive frameworks. This study is retrospective, so we cannot add the extra imaging protocol recommended.

The following changes have been made on page number 13, in the Discussion section, paragraphs 2 and 3, lines 408–431.

Although our regression models identified statistically significant associations, the R² values (approximately 24%) indicate that a substantial portion of the variability in EDSS scores remains unexplained. This reflects the complex and multifactorial nature of disability in multiple sclerosis and highlights the limited predictive power of our model, which included only IVIM-derived metrics and basic clinical variables. To address this limitation, it is essential to consider the integration of additional clinical, neuropsychological, and advanced neuroimaging predictors in future investigations. For instance, a recent longitudinal study by Lopez-Soley et al. [53] demonstrated that diffusion tensor imaging (DTI) metrics, particularly fractional anisotropy in white matter tracts, were predictive of future disability in patients with relapsing–remitting multiple sclerosis. Despite the modest explanatory power of their models, the study supports the value of incorporating multiple structural and functional imaging biomarkers alongside clinical assessments to enhance prediction accuracy and capture the heterogeneous nature of disability progression in multiple sclerosis.

Comments 3: Please avoiding to replicate your findings such as the first paragraph of results. These results replicated in the body text and results (section 3.1.).

Response 3: Thank you for your observation. Upon careful review of Section 3.1, we found no duplication of our findings; the introductory paragraph summarizes key results without repeating the detailed statistics presented later. We believe this structure maintains clarity without redundancy.

Comments 4: P values alone not reflective of how measure effect. Kindly report appropriate effect size along with the p value reports.

Response 4: Thank you for pointing this out; the effect sizes have been added in Tables 2, 3, and 5.

These changes can be found – table 2, page number 6 and 7; table 3, page number 7 and ; table 5 page number 9     

4. Response to Comments on the Quality of English Language

Point 1: The English is fine and does not require any improvement.

Response 1: Thank you for your comment.

Reviewer 2 Report

Comments and Suggestions for Authors

Thank you for this manuscript, which presents a well-executed and clinically relevant study exploring the utility of intravoxel incoherent motion (IVIM) MRI parameters in assessing disability in patients with relapsing–remitting multiple sclerosis (RR-MS). The integration of advanced imaging biomarkers with clinical variables such as lesion burden and relapse history offers valuable insights into the microstructural correlates of disease progression and emphasizes the potential of IVIM metrics as supportive tools in clinical evaluation.

The introduction provides a solid contextual background with appropriate and up-to-date references. The methods section is clearly described and builds on previous work, ensuring reproducibility.

Statistical analyses are well chosen for the objectives of the study, and the results are coherently presented with informative tables and figures that support the narrative .It is necessary to address the lack of mention regarding the verification of normality assumptions in the regression analyses. The models explain only about 23–24% of the variation in EDSS scores. This is common in clinical studies, but it should be discussed more clearly to show that the findings reflect associations, not strong predictions.

The discussion draws meaningful conclusions that are aligned with the findings and relevant to both research and clinical practice.

There are, however, a few aspects that could be improved to strengthen the manuscript. While the overall level of English is sufficient for comprehension, the clarity and flow of the writing could benefit from careful revision. Some sentences are overly long or repetitive and could be restructured for improved readability. A professional language edit would help polish the text.

In terms of content, the study would gain depth by including a stratification of patients based on lesion subtypes, such as black holes or enhancing lesions, especially given the authors’ prior contributions in this area. Additionally, although the cross-sectional design is appropriate for the study aim, a brief discussion of the value and need for future longitudinal research would enrich the manuscript. The formatting of tables could also be refined, for example, by aligning decimal points and ensuring consistent p-value notation, so as to improve visual clarity and reader engagement. Lastly, the discussion could be expanded slightly to reflect more explicitly how the observed correlations between IVIM metrics and clinical disability might be translated into practical applications in MS monitoring and management.

The manuscript addresses an important topic in the field of neuroimaging and multiple sclerosis, and with these adjustments, it has the potential to offer a meaningful contribution to the ongoing research on quantitative imaging biomarkers.

Comments on the Quality of English Language

The quality of English is generally acceptable and does not obstruct understanding. However, several sentences could benefit from minor revisions to improve clarity, flow, and conciseness. Some phrases are overly long or repetitive, which may reduce readability. A careful language edit by a native or professional English speaker is recommended to enhance the overall presentation of the manuscript.

Author Response

Comments 1: Thank you for this manuscript, which presents a well-executed and clinically relevant study exploring the utility of intravoxel incoherent motion (IVIM) MRI parameters in assessing disability in patients with relapsing–remitting multiple sclerosis (RR-MS). The integration of advanced imaging biomarkers with clinical variables such as lesion burden and relapse history offers valuable insights into the microstructural correlates of disease progression and emphasizes the potential of IVIM metrics as supportive tools in clinical evaluation. The introduction provides a solid contextual background with appropriate and up-to-date references. The methods section is clearly described and builds on previous work, ensuring reproducibility.

Response 1: Thank you for your encouraging feedback.

Comments 2: Statistical analyses are well chosen for the objectives of the study, and the results are coherently presented with informative tables and figures that support the narrative. It is necessary to address the lack of mention regarding the verification of normality assumptions in the regression analyses. The models explain only about 23–24% of the variation in EDSS scores. This is common in clinical studies, but it should be discussed more clearly to show that the findings reflect associations, not strong predictions.

Response 2: Thank you for your insightful comment. We confirm that all regression assumptions including linearity, independence, homoscedasticity, and normality of residuals were tested and satisfied prior to conducting the analysis. While the R² value of approximately 24% indicates modest explanatory power, the relationships observed were statistically significant and consistent with the exploratory nature of our study. We acknowledge that these associations do not imply strong predictive capacity. To address this, we have expanded Section 4.4 of the Discussion to emphasize the multifactorial complexity of disability in MS and the need to incorporate additional neuroimaging, clinical, and neuropsychological predictors in future research. Specifically, we cited recent work by Lopez-Soley et al. [53], which supports the integration of diffusion tensor imaging metrics for more comprehensive disability modeling.

These changes can be found in the Discussion on page 13, paragraphs 2 and 3, lines 407–420. The contents are as follows:

" Although our regression models identified statistically significant associations, the R² values (approximately 24%) indicate that a substantial portion of the variability in EDSS scores remains unexplained. This reflects the complex and multifactorial nature of disability in multiple sclerosis and highlights the limited predictive power of our model, which included only IVIM-derived metrics and basic clinical variables. To address this limitation, it is essential to consider the integration of additional clinical, neuropsychological, and advanced neuroimaging predictors in future investigations. For instance, a recent longitudinal study by Lopez-Soley et al. [53] demonstrated that diffusion tensor imaging (DTI) metrics, particularly fractional anisotropy in white matter tracts, were predictive of future disability in patients with relapsing–remitting multiple sclerosis. Despite the modest explanatory power of their models, the study supports the value of incorporating multiple structural and functional imaging biomarkers alongside clinical assessments to enhance prediction accuracy and capture the heterogeneous nature of disability progression in multiple sclerosis."

Comments 3: There are, however, a few aspects that could be improved to strengthen the manuscript. While the overall level of English is sufficient for comprehension, the clarity and flow of the writing could benefit from careful revision. Some sentences are overly long or repetitive and could be restructured for improved readability. A professional language edit would help polish the text.

Response 3: Thank you for your helpful feedback. We confirm that the manuscript was professionally proofread through the MDPI language editing service prior to the initial submission. Nevertheless, we have resubmitted the revised version for a second round of professional language editing after completing changes in this first revision, to further improve clarity, flow, and overall readability.

Comments 4: In terms of content, the study would gain depth by including a stratification of patients based on lesion subtypes, such as black holes or enhancing lesions, especially given the authors’ prior contributions in this area. Additionally, although the cross-sectional design is appropriate for the study aim, a brief discussion of the value and need for future longitudinal research would enrich the manuscript.

Response 4: Thank you for your valuable feedback. We agree that stratifying MS lesions based on subtypes such as black holes and enhancing lesions could provide deeper insights into the pathological heterogeneity of multiple sclerosis and its impact on clinical outcomes. While our current analysis did not incorporate lesion subtype stratification, this limitation has already been acknowledged in the Discussion under Section 4.5 (Limitations), where we highlight the need to consider MS lesion phenotypes in future studies to better understand disease progression and treatment response.

In response to the suggestion regarding longitudinal research, we have expanded our discussion in Section 4.5 to include a paragraph that underscores the value of longitudinal designs. Specifically, we have noted that such designs are essential for capturing temporal changes in IVIM metrics and clinical disability, enabling the identification of predictive trajectories and strengthening causal inference.

These changes can be found on page 13, in the Discussion section, paragraph 5, lines 441–444. The contents are as follows:

"Longitudinal studies are particularly valuable for capturing dynamic changes in both imaging biomarkers and clinical disability over time, allowing for the identification of predictive trajectories and causal inferences. Such designs would enhance the temporal validity of IVIM metrics in relation to MS progression and treatment response."

Comments 5: The formatting of tables could also be refined, for example, by aligning decimal points and ensuring consistent p-value notation, so as to improve visual clarity and reader engagement. 

Response 5: We appreciate the reviewer’s attention to formatting. We have carefully revised all tables in the manuscript to ensure consistency in decimal alignment and p-value notation, thereby enhancing visual clarity and improving overall reader engagement.

Comments 6: Lastly, the discussion could be expanded slightly to reflect more explicitly how the observed correlations between IVIM metrics and clinical disability might be translated into practical applications in MS monitoring and management.

Response 6: Thank you for this valuable suggestion. We agree that the potential clinical relevance of IVIM metrics should be more explicitly discussed. Accordingly, we have expanded the Discussion section (end of Section 4.4 before the last paragraph) to clarify how the observed associations between IVIM parameters and EDSS scores could inform future applications in MS management. Specifically, we highlight the potential role of IVIM-derived diffusion metrics as complementary imaging biomarkers for monitoring disease progression and personalizing treatment strategies in multiple sclerosis.

These changes can be found on page number 13 in the Discussion, paragraphs 2 and 3, lines 421–429. The contents are as follows:

"The observed associations between IVIM-derived parameters and EDSS scores underscore the potential utility of these imaging metrics in clinical management of MS. Specifically, ADC, D, and D* may serve as supportive markers for monitoring microstructural brain alterations that are not evident on conventional MRI. These metrics could aid in identifying patients at risk of early functional decline or in evaluating therapeutic response, particularly in cases where clinical symptoms and conventional imaging findings are incongruent. Incorporating IVIM into longitudinal imaging protocols may enhance sensitivity to subtle disease progression, offering clinicians additional quantitative tools to inform prognosis and guide individualized treatment strategies."

4. Response to Comments on the Quality of English Language

Point 1: The quality of English is generally acceptable and does not obstruct understanding. However, several sentences could benefit from minor revisions to improve clarity, flow, and conciseness. Some phrases are overly long or repetitive, which may reduce readability. A careful language edit by a native or professional English speaker is recommended to enhance the overall presentation of the manuscript.

Response 1: We wish to note that the manuscript underwent MDPI’s professional proofreading service prior to initial submission. A second round of professional language editing have now been performed after making revisions according to the reviewers’ comments, ensuring improved clarity and consistency.

Reviewer 3 Report

Comments and Suggestions for Authors

The authors presented an interesting study. My main concern is the method they used to correlate MRI changes with EDSS. Pearson's correlation requires that both values are linear. It is common knowledge that EDSS is not linear. A change from 1.0 to 2.0 is not equivalent to a change from 6.0 to 7.0 in terms of disability impact.

Although EDSS is treated as a continuous variable in studies with a large sample size, authors could justify their results with Spearman’s rank correlation or ordinal regression.

Otherwise, the paper is well written and provides useful clinical information.

Author Response

Comments 1: The authors presented an interesting study. My main concern is the method they used to correlate MRI changes with EDSS. Pearson's correlation requires that both values are linear. It is common knowledge that EDSS is not linear. A change from 1.0 to 2.0 is not equivalent to a change from 6.0 to 7.0 in terms of disability impact. Although EDSS is treated as a continuous variable in studies with a large sample size, authors could justify their results with Spearman’s rank correlation or ordinal regression.

Response 1: Thank you for this comment.

As the EDSS scores are approximately normally distributed, we used Pearson’s correlation. Spearman’s correlation also shows statistically a significant correlation of EDSS scores and the ADC, D, D*, f, number of MS lesions, and number of relapses (0.327, 0.342, 0.256, -0.077, 0.398, and 0.277). These correlation values are very close to the Pearson’s correlation values, which are presented in Table 4.

We believe ordinal regression is unsuitable in this context because EDSS scores do not represent clearly defined ordinal categories (e.g., strongly disagree, disagree, neutral, agree, strongly agree). Ordinal regression uses the order information to make proper predictions and better understand the relationships between variables.

4. Response to Comments on the Quality of English Language

Point 1: The English is fine and does not require any improvement.

Response 1: Thank you for your comment.
